

# Using solar radiation data in soil moisture diagnostic equation for estimating root-zone soil moisture

Olumide Omotere[1], Feifei Pan[2] and Lei Wang[1]

[1] Geography & Anthropology, Louisiana State University and Agricultural and Mechanical College, Baton Rouge, Louisiana, USA
[2] Geography and The Environment, University of North Texas, Denton, Texas, USA

## ABSTRACT

The soil moisture daily diagnostic equation (SMDE) evaluates the relationship between the loss function coefficient and the summation of the weighted average of precipitation. The loss function coefficient uses the day of the year (DOY) to approximate the seasonal changes in soil moisture loss for a given location. Solar radiation is the source of the energy that drives the complex and intricates of the earth-atmospheric processes and biogeochemical cycles in the environment. Previous research assumed DOY is the approximation of other environmental factors (*e.g.*, temperature, wind speed, solar radiation). In this article, two solar radiation parameters were introduced, *i.e.*, the actual solar radiation and the clear sky solar radiation and were incorporated into the loss function coefficient to improve its estimation. This was applied to 2 years of continuous rainfall, soil moisture data from USDA soil climate network (SCAN) sites AL2053, GA2027 MS2025, and TN2076. It was observed that the correlation coefficient between the observed soil moisture and B values (which is the cumulated average of rainfall to soil moisture loss) increased on average by 2.3% and the root mean square errors (RMSEs) for estimating volumetric soil moisture at columns 0–5, 0–10, 0–20, 0–50, 0–100 cm reduced on average by 8.6% for all the study sites. The study has confirmed that using actual solar radiation data in the soil moisture daily diagnostic equation can improve its accuracy.

# INTRODUCTION

A soil profile can be divided into saturated and unsaturated zones. Soil moisture is the water found in the unsaturated zone, which is available for plant usage (*Hornberger et al., 2014*). The main source of moisture for the unsaturated zone is from precipitation through the infiltration process, and moisture is lost by evaporation from a bare surface and evapotranspiration from a vegetated surface. The vertical flow of soil moisture eventually recharges the water table leading to moisture lost in the unsaturated zone (*Ridolfi et al., 2003*). The significance of soil moisture in the environment is numerous; for example, soil moisture is responsible for the segmentation of surface runoff, infiltration process, and percolation of water in the hydrological cycle. Soil moisture governs the energy interaction

Corresponding author
Lei Wang, leiwang@lsu.edu

from the land surface to the atmosphere because it regulates sensible and latent heat fluxes (*Entekhabi & Rodriguez-Iturbe, 1994*; *D'Odorico & Porporato, 2004*).

The zone where the plant root is situated is called the plant root zone. The growth of a plant depends on the condition of the plant root zone and the moisture content of the soil (*Hanson, Rojas & Shaffer, 1999*). Plants manufacture their food through the process of photosynthesis, using soil moisture, carbon dioxide, and sunlight. Soil moisture dries down as the rate of photosynthesis and evapotranspiration increases (*e.g.*, *Nie et al., 1992*). Soil moisture dynamics is important in understanding the complexity of environmental processes (*Pan, Peters-Lidard & Sale, 2003*). Solar radiation and heat fluxes are impacted by root zone soil moisture variation (*Entekhabi, Rodriguez-Iturbe & Castelli, 1996*). The partition between the atmospheric turbulent influx and the thermal fluxes caused by the land surface temperature and the atmospheric stability near the surface (*Entekhabi, Rodriguez-Iturbe & Castelli, 1996*; *Settin et al., 2007*; *Khong et al., 2015*). This partition is dependent on soil moisture changes (*Zhu & Lin, 2011*).

The simplification of the linear stochastic water balance equation (*Entekhabi & Rodriguez-Iturbe, 1994*) leads to the soil moisture daily diagnostic equation (SMDE) (*Pan, Peters-Lidard & Sale, 2003*; *Pan, 2012*; *Pan & Nieswiadomy, 2016*). The soil moisture diagnostic equation is based on the ratio of estimated soil moisture loss to the weighted summation of historical rainfall (*Pan, Peters-Lidard & Sale, 2003*; *Pan, 2012*; *Pan & Nieswiadomy, 2016*) for soil moisture estimation. The soil moisture diagnostic equation is easy to apply to agricultural and irrigation practices because it is uncomplicated for the estimation of soil moisture dynamics. Information about the antecedent soil moisture is not required (*Pan, 2012*; *Pan, Nieswiadomy & Qian, 2015*). While the direct measurement method requires recalibration to eliminate cumulative errors, the SMDE does not require any form of recalibration. With the advent of advanced computing many studies have explored other approaches for estimating soil moisture with reasonable accuracy, machine learning, remote sensing, google earth engine offers unpredicted approach to evaluating temporal and spatial pattern of soil moisture estimation (*Greifeneder, Notarnicola & Wagner, 2021*; *Kisekka et al., 2022*; *Ahmad, Forman & Kumar, 2021*). However, most machine learning methods depend on large dataset for accuracy, and most global monitoring satellites of soil moisture are limited to surface soil moisture (*Scowen et al., 2021*).

Since evapotranspiration is correlated positively with the air temperature and solar radiation (*e.g.*, *Thornthwaite, 1948*; *Monteith, 1965*; *Priestley & Taylor, 1972*), other environmental variables can enhance the estimation of the loss function coefficient (*Pan, Peters-Lidard & Sale, 2003*; *Pan, 2012*; *Pan & Nieswiadomy, 2016*). This study examines the effect of solar radiation in estimating soil moisture using SMDE. The objective of this study is to improve the daily diagnostic equation by integrating the ratio of actual solar radiation to the long-term average of solar radiation or clear sky solar radiation into the loss function coefficient of the daily diagnostic equation to predict the root-zone soil moisture. The sinusoidal wave function is used in the soil moisture loss function to imitate the seasonal variation of soil moisture loss or to show moisture dry down through a year. The current soil moisture loss function in the SMDE uses the day of the year (DOY) to approximate soil

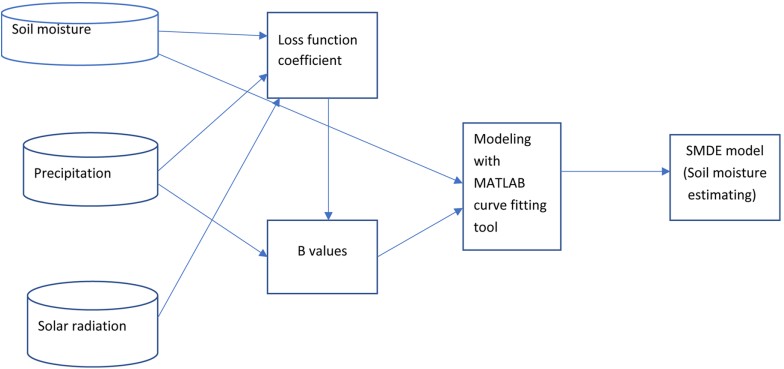

**Figure 1 Flow chart for the soil moisture estimation using SMDE.**

moisture loss because it varies with seasons (*Pan, Peters-Lidard & Sale, 2003*; *Pan, 2012*; *Pan, Nieswiadomy & Qian, 2015*; *Pan & Nieswiadomy, 2016*), but soil moisture loss depends on other environmental factors. The seasonal dynamic of solar radiation mimics this same pattern (summer, high and winter, low). However, the evaporation of soil water content driven by solar radiation was not considered in the equation. The energy and water exchanges between the atmosphere and the vegetation are propelled by solar insolation (*i.e.*, evaporation and evapotranspiration) (*Entekhabi, Rodriguez-Iturbe & Castelli, 1996*; *Maeda et al., 2017*), hence affecting soil moisture dry-down. Clear sky solar radiation is the quantity of solar radiation reaching the surface of the earth without absorption, reflection, and scattering by cloud, dust, aerosol, and water vapor in the atmosphere ($R_{cs}$) (*Larrañeta et al., 2017*). Conversely, the actual solar radiation ($R_{as}$) is the radiation reaching the earth's surface after the interference of clouds and other particles in the atmosphere.

We introduced solar radiation parameters because it is the driving force in nature for all biogeochemical cycles, and its changes mimic the soil moisture loss in the studied locations. The new model can account for daily weather patterns such as cloudy days and clear sky days, which influences the amount of solar radiation that is received on a given day. The objective of this research article is to make provision for this pattern, thus improving the estimation of soil moisture loss. This research examines the tradeoff between the application of the long-term average actual solar radiation to the loss function coefficient of the SMDE. The specific research question and contribution of this research is the quantification of the improvement in the accuracy of the root-zone soil moisture estimation after incorporating clear sky solar radiation and the actual solar radiation into the loss function coefficient of the daily diagnostic equation. Figure 1 depicts the flowchart of the processes implemented in the research.

## MATERIAL AND METHOD

### Soil moisture diagnostic equation

The soil moisture stochastic differential equation (*Entekhabi & Rodriguez-Iturbe, 1994*) was simplified into the soil moisture diagnostic equation (*Pan, Peters-Lidard & Sale, 2003*; *Pan, 2012*; *Pan, Nieswiadomy & Qian, 2015*; *Pan & Nieswiadomy, 2016*).

$$k\frac{d\theta}{dp} = -\eta\theta + \gamma M \tag{1}$$

where k is the length of the soil column, from the surface to depth k, $\theta$ is the soil moisture in column k, p is the time, the soil moisture loss coefficient is denoted by $\eta$, M is the rate of precipitation, and the coefficient of infiltration is $\gamma$. The equation explores evapotranspiration, evaporation, and drainage to explain soil moisture loss. It is simplified into the soil moisture diagnostic equation (*Pan, Peters-Lidard & Sale, 2003*; *Pan, 2012*; *Pan, Nieswiadomy & Qian, 2015*). The detail about the derivation is explained below. Equation (1) was modified into Eq. (2).

$$dp = \frac{kd\theta}{-\eta\theta + \gamma M} \tag{2}$$

The soil moisture was observed as a time series at a particular point, Eq. (2) is integrated between $P_2$ and $P_1$.

$$\int_{P_2}^{P_1} \frac{kd\theta}{-\eta\theta + \gamma M} = \int_{P_2}^{P_1} dp \tag{3}$$

It is assumed that the infiltration coefficient and the loss coefficient for a short time step ($\leq 1$ day) are independent of time. The observed rainfall between time $P_1$ and $P_2$ is M, and it is independent of soil moisture.

$$-\frac{k}{\eta_1} \ln\left[\frac{\theta_1 - \frac{\gamma M_1}{\eta_1}}{\theta_2 - \frac{\gamma M_1}{\eta_1}}\right] = P_1 - P_2 \tag{4}$$

The $\eta_1$ and $M_1$ are the loss coefficient and the cumulative precipitation between time $P_1$ and $P_2$ respectively. Equation (4) was modified into Eq. (5).

$$\theta_1 = \theta_2 e^{-\frac{\eta_1}{k}(P_1 - P_2)} + \frac{\gamma M_1}{\eta_1}\left[1 - e^{-\frac{\eta_1}{k}(P_1 - P_2)}\right] \tag{5}$$

To show the daily time step (*i.e.*, $P_1 - P_2 = 1$ day), Eq. (5) was simplified into Eq. (6a).

$$\theta_1 = \theta_2 e^{-\frac{\eta_1}{k}} + \frac{\gamma M_1}{\eta_1}\left[1 - e^{-\frac{\eta_1}{k}}\right] \tag{6a}$$

Here, $\eta_1$, $\theta_1$, and $M_1$ are the daily loss coefficient, soil moisture, and precipitation for day 1, $\theta_2$ = soil moisture on Day 2. Day 1 is behind day 2. This can also be expressed as

$$\theta_2 = \theta_3 e^{-\frac{\eta_2}{k}} + \frac{\gamma M_2}{\eta_2}\left[1 - e^{-\frac{\eta_2}{k}}\right] \tag{6b}$$

$$\theta_{n-1} = \theta_n e^{-\frac{\eta_{n-1}}{k}} + \frac{\gamma M_{n-1}}{\eta_{n-1}}\left[1 - e^{-\frac{\eta_{n-1}}{k}}\right] \tag{6c}$$

Equations (6b) and (6c) were substituted into Eq. (6a), leading to Eq. (7):

$$\theta_1 = \theta_n e^{-\sum_{i=1}^{i=n-1}(\eta_1/k)} + \sum_{i=2}^{i=n-1}\left[\frac{\gamma M_i}{\eta_i}(1-e^{-\frac{\eta_i}{k}})e^{-\sum_{j=1}^{j=i-1}(\eta_j/k)}\right] + \frac{\gamma M_1}{\eta_1}\left(1-e^{-\frac{\eta_1}{k}}\right) \tag{7}$$

The exponential term $-\left[\sum_{n-1}^{i=n-1}(\eta/k)\right]$ approaches zero in Eq. (7) as the window size (*i.e.*, n) increases, and this reduces the leading term in Eq. (7). Consequently, the soil moisture can be calculated directly from the cumulative average rainfall without data from the initial soil moisture at threshold time window size n (*Pan, 2012*).

$$\theta_1 = \sum_{i=2}^{i=n-1}\left[\frac{\gamma M_i}{\eta_i}(1-e^{-\frac{\eta_i}{k}})e^{-\sum_{j=1}^{j=i-1}(\eta_j/k)}\right] + \frac{\gamma M_1}{\eta_1}\left(1-e^{-\frac{\eta_1}{k}}\right) = \gamma B \tag{8}$$

The B in Eq. (8) is defined below in Eq. (9),

$$B = \sum_{i=2}^{i=n-1}\left[\frac{M_i}{\eta_i}(1-e^{-\frac{\eta_i}{k}})e^{-\sum_{j=1}^{j=i-1}(\eta_j/k)}\right] + \frac{M_1}{\eta_1}\left(1-e^{-\frac{\eta_1}{k}}\right) \tag{9}$$

The B value is defined as the summation of the weighted ratio of rainfall rate to loss coefficient (*Pan, 2012*), The rainfall contribution to soil moisture from day 1 diminishes due to the decreasing exponential term $-\left[\sum_{n-1}^{i=n-1}(\eta/k)\right]$ in Eq. (8) (*Pan, 2012*).

## Relationship between soil moisture and B value

The derivation of the soil moisture diagnostic equation shows the importance of determining the infiltration rate. The rate of infiltration varies with the soil moisture content (*Pan, 2012*). As the infiltration rate diminishes and consequently becomes zero, the B value increases, and the value of the soil moisture increases and becomes constant (*Pan, Peters-Lidard & Sale, 2003*; *Pan, 2012*). The best expression for soil moisture is a function of the B values denoted by Eq. (10) (*Pan, Peters-Lidard & Sale, 2003*; *Pan, 2012*).

$$\theta = \theta_{re} + (\phi_e - \theta_{re})\left(1 - e^{-C_4 B}\right) \tag{10}$$

where $\emptyset_e$, $\theta_{re}$, and $C_4$ denote effective soil porosity, the effective residual soil moisture, and soil hydraulic properties, respectively. They are computed using the MATLAB curve fitting tool to fit the observed soil moisture and the B values.

## Improved soil moisture loss coefficient

To estimate the soil moisture, the soil moisture loss coefficient function is required and is dependent on the evapotranspiration and drainage. These are the factors controlling the rate of soil moisture loss coefficient (*Pan, 2012*). The temporal dynamics of evapotranspiration are most strongly correlated with seasons, *i.e.*, higher evapotranspiration in summer and lower evapotranspiration in winter (*Pan, 2012*; *Pan, Nieswiadomy & Qian, 2015*; *Pan & Nieswiadomy, 2016*). The loss function coefficient is based on an approximation of all the seasonal factors to days of the year (DOY) (*Pan, 2012*). *Pan, Nieswiadomy & Qian (2015)* examined the comparison of loss function

coefficient (Eq. (12)) to the linear function of potential evapotranspiration (Eq. (11)) and concluded that the loss coefficient function (sinusoidal wave function) is slightly better at estimating soil moisture.

$$\eta = a + b \times PET \tag{11}$$

$$\eta_i = C_1 + C_2 \; \sin\left[\frac{2\pi(DOY_i + C_3)}{365}\right] \tag{12}$$

$\eta$ denote the loss coefficient for the day i, C1, and C2 and C3 denote the parameters for loss function coefficient and the DOY is the day of the year. C1, C2, and C3 denote the mean, magnitude, and phase of sinusoidal wave function respectively and in Eq. (11), a and b are positive constants (*Pan, 2012*; *Pan, Nieswiadomy & Qian, 2015*; *Pan & Nieswiadomy, 2016*).

The improved soil moisture loss coefficient (Eq. (13)) incorporates solar radiation parameters into the soil moisture diagnostic equation for estimating soil moisture. This is based on the premise that the soil moisture loss coefficient mimics the seasonal variation of solar radiation (summer, high and winter, low). The new parameters introduced into the loss function coefficient are the clear sky solar radiation ($r_{ci}$) (the long-term average daily mean solar radiation) and the actual solar radiation($r_{ai}$) to account for the effect of cloud cover. The distinct characteristic of clear sky solar radiation is that it increases monotonously from the first day of the year until it gets to the maximum on the 172[nd] day of the year in the northern hemisphere, and it starts to decrease monotonously until the last day of the year. To estimate the actual solar radiation, this article uses 5 years of solar radiation data. The estimation is based on calculating the daily maximum solar radiation for each day (*e.g.*, maximum January 1 for 5 years) of the year for 5 years using MATLAB coding (Fig. 2). The daily maximum solar radiation was used to estimate the clear sky radiation. MATLAB curve fitting tool was used to fit the monotonously selected data to compute the clear solar ($r_{ci}$) radiation. The solar radiation parameters were integrated into the loss function coefficient (Eq. (13)) to improve the accuracy of soil moisture estimates.

$$\eta_i = C_1 + C_2 \left( \sin\lfloor\frac{2\pi(DOY_i + C_3)}{365}\rfloor \lfloor\frac{r_{ai}}{r_{ci}}\rfloor \right) \tag{13}$$

## Determination of loss function parameters

The loss coefficient parameters ($C_1$, $C_2$, and $C_3$) are highly correlated with nature of the surface and geographic location (*Pan, 2012*; *Pan, Nieswiadomy & Qian, 2015*). These parameters were estimated by maximizing the coefficient of correlation between the B values and the observed soil moisture (*Pan, 2012*). These parameters are not negative numbers; $C_1$ and $C_2$ have the same unit as the precipitation and $C_3$ as the DOY. The range recommended for $C_1$ is 0–4 cm/day in mid-latitude regions (*Pan, Nieswiadomy & Qian, 2015*; *Pan & Nieswiadomy, 2016*), and this research employs the same approach because the study areas are in the mid-latitude. It is assumed that the infiltration coefficient and the loss coefficient for a short time step ($\leq$1 day) are independent of time.
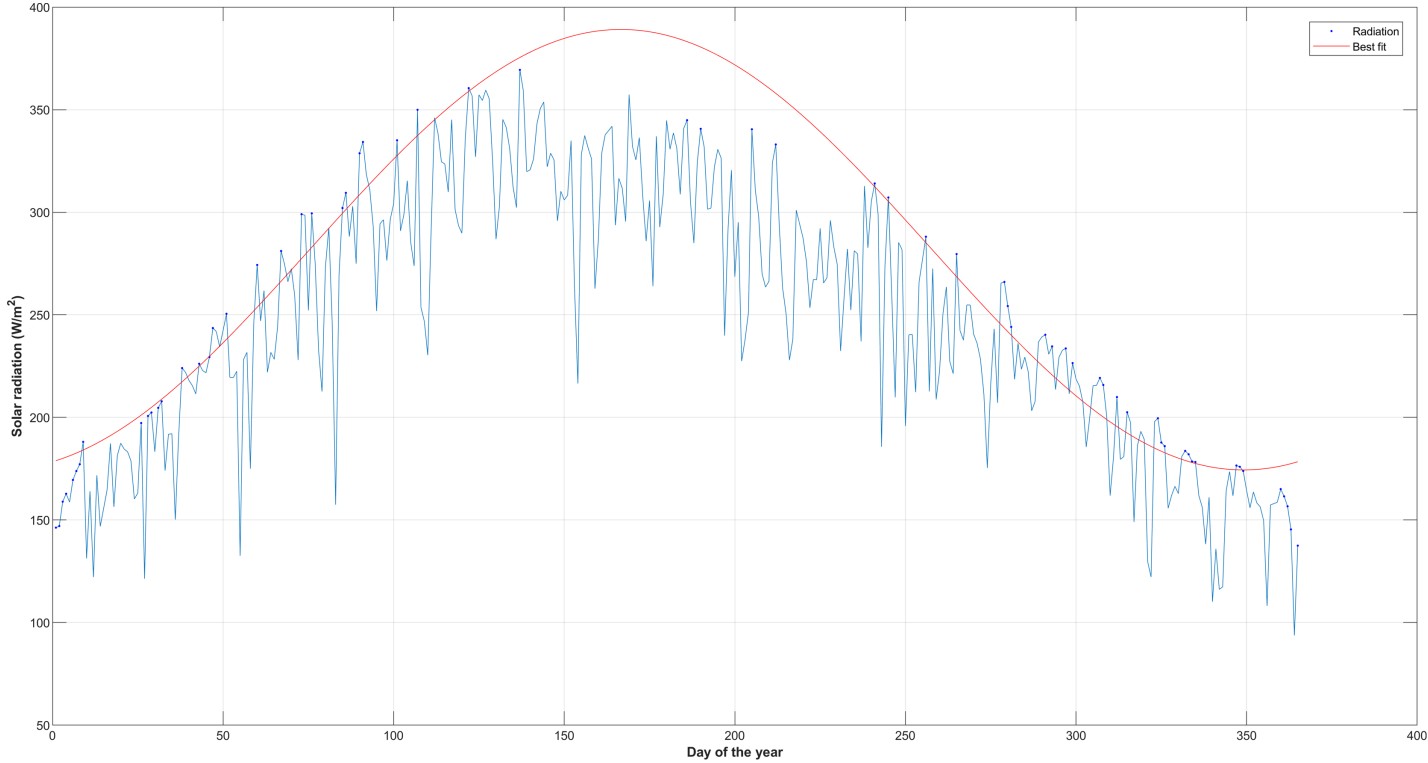

**Figure 2  Maximum actual solar radiation for each day of the year for 5 years for TN2076.**

$$\max\left\{\frac{\sum_{i=1}^{m}\left[(\theta_i - \bar{\theta})(B_i - \bar{B})\right]}{\sqrt{\sum_{i=1}^{m}(\theta_i - \theta)^{-2}}\sqrt{\sum_{i=1}^{n}(B_i - \bar{B})^2}}\right\}\Bigg|_{C_1, C_2, C_3} \tag{14}$$

$\theta$ and B are the soil moisture measurement and the computed B values, where i = 1, 2, 3……………. n. The global search technique was applied to maximize the loss function in this article (*Pan, 2012*; *Pan, Nieswiadomy & Qian, 2015*; *Pan & Nieswiadomy, 2016*). The global search is lucid (*Pan, Nieswiadomy & Qian, 2015*), and the searching domain is expressed below in Eq. (15).

Searching domain $= \{0 < C1 < 4\,\text{cm}/\text{day}; 0 < C2 \leq C1; 0 < c3 < 366\} \tag{15}$

## STUDY AREA AND DATA

The research data is from the United States National Water and Climate Center, the National Resources Conservation Services, the National Water and Climatic Center (NWCC), and the Soil Climatic Network. The observed soil moisture data at various depths (5, 10, 20, 50 and 100 cm) can be downloaded from https://www.wcc.nrcs.usda.gov/webmap. The research utilizes 5 years of downloaded actual solar radiation, 2 years of soil moisture, and rainfall data from sites in Alabama, Georgia, Tennessee, and Mississippi. The characteristics of the study areas are described in Table 1. The soil moisture content in the root zone can be determined using the improved daily diagnostic equation for soil

**Table 1 Examined SCAN sites.**

| Site ID | State | Lat./Long. | Landcover | Soil texture | Parameter period | Model period |
|---------|-------|-----------|-----------|--------------|------------------|--------------|
| AL2053 | Alabama | 34.9°N–86.53°W | Grass | Silt clay | 1/1/14–31/12/15 | 1/1/17–31/12/17 |
| GA2027 | Georgia | 31.30°N–83.33°W | Bare | Loam | 1/1/14–31/12/15 | 1/1/16–31/12/16 |
| MS2025 | Mississippi | 34.23°N–89.9°W | Bare | Silt | 1/1/15–31/12/16 | 1/1/17–31/12/17 |
| TN2076 | Tennessee | 35.07°N–86.89°W | Grass | Silt | 1/1/13–31/12/14 | 1/1/18–31/12/18 |

moisture estimation. The soil moisture content for the entire column was calculated using Eq. (16).

$$\theta = \frac{\theta_1 k_1 + \sum_{i=2}^{n} [(k_i - k_{i-1})(\theta_i + \theta_{i-1})/2]}{k_n} \tag{16}$$

$\theta_i$ denotes the soil moisture content at depth $k_i$, i denotes 1......... n and the average soil moisture in the entire root zone is denoted by $\theta$.

The site GA2027 is Tifton loamy sand, which is 24.6 percent clay, 66.6 percent sand, 8.8 percent silt, pH is 5.4, and loamy sand texture. The profile from 0 to 11 inches is loamy sand, 11 to 22 inches is fine sandy loam, 22 to 40 inches is sandy clay loam, 40 to 50 inches is sandy clay loam, 50 to 60 inches is paragravelly sandy clay loam, 60 to 65 inches is sandy clay, 65 to 80 inches is sandy clay loam. The soil in AL2053 is Cookeville silt loam, and the clay is 37.3 percent, 21.0 percent sand and silt are 41.6 percent, pH is 5.1, and the texture silt loamy. The profile ranges from 0 to 8 inches in silt loam, 8 to 28 inches in silty clay loam, and 28 to 80 inches in gravelly clay. MS2025 Gullied land, silty, clay is 19.4 percent, 11.5 percent sand, silt is 69.0 percent, the pH is 5.3, and the texture is silt loam. The profile is 0 to 9 inches is silt loam, 9 to 23 inches is silty clay loam, and 23 to 80 inches is silt loam. TN2076 soil Braxton cherty silt loam, clay is 47.9 percent, 22 percent sand, and silt is 30.2, and the is 5.6, and the texture is gravelly silt loam, the profile from 0 to 5 inches is gravelly silt loam, 5 to 60 inches is clay.

## RESULTS

We generate the loss coefficient parameters ($C_1$, $C_2$, and $C_3$) using the Monte Carlo search method by maximizing the coefficient of correlation between the B values and the observed soil moisture for each column (Table 2). The correlation coefficient ($R^2_{\theta B}$) between the observed soil moisture and the B values at columns 0–5, 0–10, 0–20, 0–50, 0–100 cm, shows improvement in soil moisture estimation for the entire root zone.

The inclusion of solar radiation was studied by comparing the original SMDE (without solar radiation data) to SMDE with solar radiation in Table 2 which shows the improved SMDE has a good prediction for the studied sites. The correlation ($R^2_{\theta B}$) results show 0–5 cm columns for all the sites improved by 0.02 on average, for 0–10 cm columns by 0.01, 0–20 cm columns by 0.01. This was achieved after incorporating the ratio of actual solar radiation to clear sky solar radiation into the loss function coefficient.

There were some exceptions in these results after the introduction of solar radiation data into loss function coefficient at 0–20 cm columns and 0–50 cm columns, where the

**Table 2 Comparing correlation coefficient and loss coefficient parameters of the improved loss coefficient function and the current loss function coefficient.**

**0–5 cm**

| | Soil moisture estimation with solar radiation | | | | Soil moisture estimation without solar radiation | | | |
|---|---|---|---|---|---|---|---|---|
| Site ID | C1 | C2 | C3 | $R^2_{B,\theta}$ | C1 | C2 | C3 | $R^2_{B,\theta}$ |
| AL2053 | 0.218 | 0.1708 | 259.913 | 0.7096 | 0.2874 | 0.2019 | 247.875 | 0.7075 |
| GA2027 | 1.1887 | 0.4704 | 52.2337 | 0.7219 | 1.1102 | 0.3177 | 42.3304 | 0.6553 |
| MS2025 | 0.2124 | 0.1357 | 249.532 | 0.917 | 0.2195 | 0.0663 | 250.859 | 0.9154 |
| TN2076 | 1.0516 | 0.7825 | 237.066 | 0.844 | 0.7583 | 0.5475 | 241.199 | 0.8431 |
| **0–10 cm** | | | | | | | | |
| AL2053 | 0.4614 | 0.4276 | 236.18 | 0.7849 | 0.4418 | 0.2096 | 235.644 | 0.7684 |
| GA2027 | 2.4080 | 0.7732 | 83.3 | 0.7802 | 1.7973 | 0.3384 | 74.1578 | 0.749 |
| MS2025 | 0.3700 | 0.1746 | 252.069 | 0.8976 | 0.4154 | 0.1319 | 236.752 | 0.8965 |
| TN2076 | 1.0751 | 0.9961 | 243.792 | 0.896 | 0.9422 | 0.5949 | 240.148 | 0.8919 |
| **0–20 cm** | | | | | | | | |
| AL2053 | 1.9899 | 1.8382 | 230.47 | 0.8597 | 3.4755 | 1.0607 | 225.466 | 0.8615 |
| GA2027 | 3.3939 | 0.9705 | 234.76 | 0.8221 | 3.6283 | 0.7528 | 232.2 | 0.823 |
| MS2025 | 0.7552 | 0.2927 | 243.771 | 0.8942 | 0.7092 | 0.1889 | 243.802 | 0.8957 |
| TN2076 | 1.4823 | 1.3524 | 255.934 | 0.7867 | 1.621 | 0.9164 | 250.658 | 0.7755 |
| **0–50 cm** | | | | | | | | |
| GA2027 | 3.9886 | 0.2145 | 145.078 | 0.8237 | 3.9976 | 0.1746 | 259.248 | 0.8374 |
| MS2025 | 1.4813 | 1.125 | 241.478 | 0.9505 | 0.6006 | 0.2626 | 238.212 | 0.9513 |
| TN2076 | 3.7561 | 1.8567 | 273.574 | 0.7696 | 2.7713 | 1.3741 | 270.876 | 0.7824 |
| AL2053 (0–100 cm) | 1.8743 | 1.873 | 243.955 | 0.8511 | 1.443 | 1.1018 | 242.868 | 0.8397 |

model did not improve after incorporating solar radiation. These exceptions may be due to the limited length of the solar radiation data available to estimated clear-sky solar radiation. To calculate the clear sky solar radiation, 10 years of actual solar radiation is recommended, but all the sites studied only had 5 years of continuous actual solar radiation data. Within 5 years of observation, not all sites have the cloud-free condition for every day of the 365-day cycle.

The three parameters of the daily diagnostic equation for estimating the root zone soil moisture (SMDE) are the effective soil porosity ($\varnothing_{re}$), effective residual soil moisture($\theta_{re}$), and empirical constant for soil hydraulic properties($C_4$). These three parameters were determined by using the MATLAB curve fitting tool to find the line of best fit (Fig. 3) for the plot of the observed soil moisture *vs* B values for each column (0–5, 0–10, 0–20, 0–50, and 0–100 cm) using the least square method (the diagnostic equation (Eq. (10))) was used as the fitting equation (Table 3) (*Pan, 2012*). The root mean square errors (RMSEs) are generally below 4.5%v/v for the entire root zone (compare to (*Pan, 2012*; *Pan, Nieswiadomy & Qian, 2015*)), which was below 5%v/v before applying the ratio of actual solar radiation to clear sky solar radiation into the loss function coefficient of the soil moisture diagnostic equation (SMDE). The decrease in the RMSEs (Table 3) further

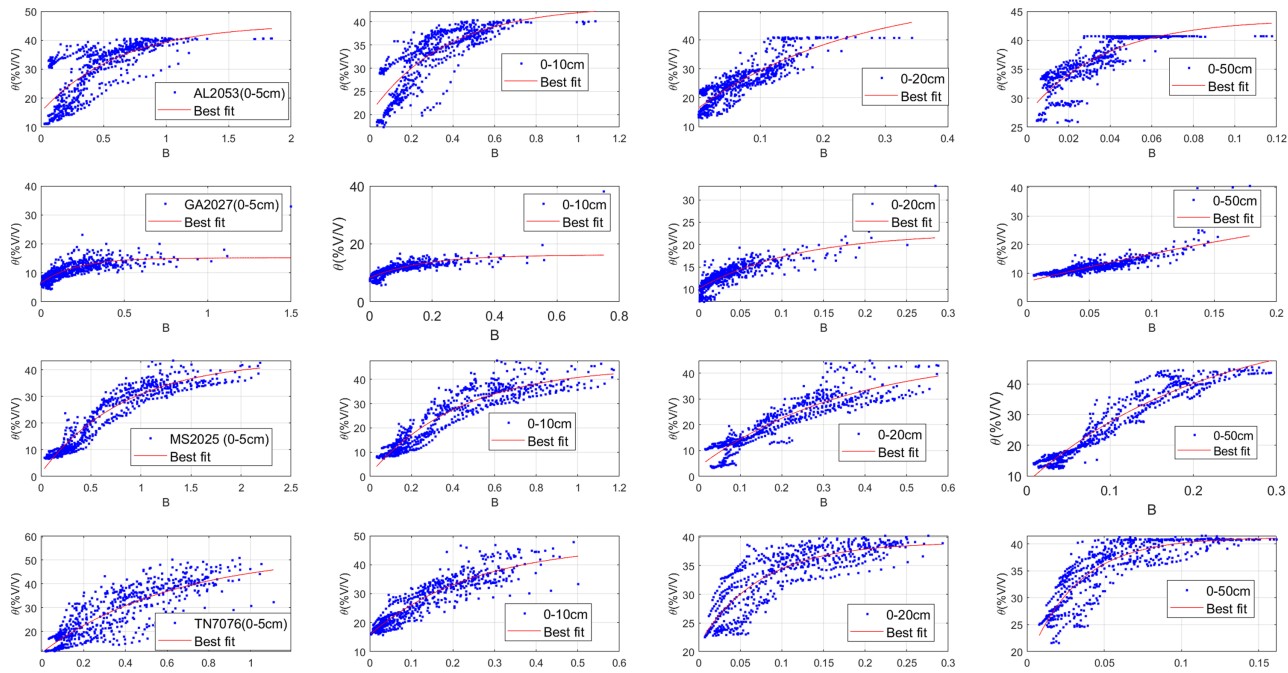

**Figure 3 The scatterplots with the line of best fit for observed soil moisture *vs* the B values for columns 0–5, 0–10, 0–20, 0–50 0–100 cm for parameter testing periods.**

indicates an improvement in the soil moisture diagnostic equation in estimating root-zone soil moisture content after incorporating solar radiation (Fig. 4).

For columns 0–5 cm, all the sites experience reduced RMSE in the experiment. The RMSE for TN2076 was reduced by 44.51 percent, the highest improvement. The slightest reduction in RMSEs was observed at the GA2027 with a 0.05 percent reduction. At columns 0–10 cm, all the RMSEs reduce after applying radiation into the loss coefficient function at columns 0–5 cm. AL2053 has the highest reduction in RMSE of 3.27 percent, and TN2076 has a 0.024 percentage decrease in the RMSE. Colum 0–20 cm TN2076 and GA2027 consistently experience a reduction in their RMSE by an average of 0.02. The RMSE in columns 0–50 cm for GA2027 and TN2076 reduce on average by 2.4 percent (Table 3). The correlation coefficient between the observed and estimated soil moisture (Table 4) is significant for the entire column (Fig. 5) for the method testing period.

## DISCUSSION

Four sites were examined for this article, spanning through the southeastern states of the contiguous U.S. It was observed that RMSEs and correlation varied spatially for all the sites examined. The correlation between the observed soil moisture and the estimated soil moisture (Table 3) increased by average 6.7% for 0–5 cm and 0.86% for the 0–10 cm, but the increase was not consistent for 0–20 and 0–50 cm. The improvement in the estimation was higher and consistent in the column 0–5 and 0–10 cm for the all the sites, but beyond these depths (*e.g.*, 0–20, 0–50 and 0–100 cm) there were some exceptions. (Tables 2 and 3). Their RMSEs for the observed soil and B values range from 1.94%v/v to 5.90%(v/v) for

**Table 3 Comparing optimal soil moisture diagnostic equation parameters and the root mean square errors.**

**0–5 cm**

| Site ID | Soil moisture estimation with solar radiation | | | | Soil moisture estimation without solar radiation | | | |
|---|---|---|---|---|---|---|---|---|
| | $\theta_{re}$ | $\emptyset_e$ | $C_4$ | RMSE | $\theta_{re}$ | $\emptyset_e$ | $C_4$ | RMSE |
| AL2053 | 11.31 | 40.45 | 2.705 | 4.145 | 15.99 | 43.15 | 1.49 | 5.939 |
| GA2027 | 6.497 | 15.17 | 4.763 | 1.95 | 6.859 | 14.61 | 11.13 | 2.051 |
| MS2025 | 1.07 | 43.85 | 1.008 | 3.224 | 0.774 | 44.19 | 1.143 | 3.272 |
| TN2076 | 14.43 | 45.04 | 2.763 | 2.793 | 14.04 | 46.26 | 3.31 | 5.033 |
| **0–10 cm** | | | | | | | | |
| AL2053 | 19.46 | 42.44 | 3.129 | 3.436 | 18.84 | 42.76 | 2.009 | 3.552 |
| GA2027 | 8.298 | 16.12 | 5.953 | 1.449 | 7.963 | 15.15 | 11.1 | 1.474 |
| MS2025 | 1.265 | 45.4 | 2.363 | 3.918 | 1.355 | 45.45 | 2.33 | 3.904 |
| TN2076 | 16.06 | 47.52 | 3.73 | 3.164 | 15.67 | 46.15 | 3.982 | 3.188 |
| **0–20 cm** | | | | | | | | |
| AL2053 | 16.36 | 56.38 | 3.944 | 3.467 | 16.21 | 52.88 | 4.695 | 3.369 |
| GA2027 | 10.14 | 22.64 | 8.418 | 1.473 | 10.22 | 22.65 | 8.451 | 1.478 |
| MS2025 | 3.692 | 47.64 | 2.808 | 3.989 | 3.506 | 47.24 | 2.651 | 3.965 |
| TN2076 | 21.26 | 39.05 | 13.52 | 2.741 | 22.85 | 39.5 | 12.68 | 2.879 |
| **0–50 cm** | | | | | | | | |
| GA2027 | 7.264 | 97.7 | 1.11 | 1.983 | 7.686 | 32.81 | 6.77 | 2.056 |
| MS2025 | 7.919 | 61.57 | 4.606 | 2.964 | 7.494 | 61.14 | 1.859 | 2.912 |
| TN2076 | 17.89 | 41.18 | 31.06 | 2.652 | 19.03 | 41.47 | 34.53 | 2.687 |
| AL2053 (0–100 cm) | 27.22 | 43.64 | 27.07 | 2.041 | 20.88 | 43.77 | 18.29 | 1.896 |

0–5 cm columns. The same pattern was observed in 0–10 and 0–20 cm columns. A total of 0–50 cm column RMSEs for all the sites were between 1.5%(v/v) and 5.90%(v/v) (Table 3). The RMSEs for sites in Georgia and Mississippi ranged from 1.5%(v/v) to 3.6%(v/v) and, for Tennessee and Alabama, 2.6%(v/v) to 5.9%(v/v).

This spatial differential pattern in the RMSEs in estimating soil moisture is a result of the prevalent environmental factors in each of the sites. With the introduction of solar radiation, we observed an improvement in the RMSEs for all the sites. The errors observed in the sites that have vegetation were higher compared to sites without vegetation. We speculate that some vegetation information such as leaf area index parameters for these sites might improve the accuracy. In fact, the rate of evapotranspiration is dependent on the leaf area index (*Wiegand, Richardson & Kanemasu, 1979*). In evaluating TN2076 and AL2053 (with vegetation), we observed the RMSEs ranged from 2.7%(v/v) to 5.9%(v/v) for all columns. This pattern is different from other sites *e.g.*, GA2027 and MS2025 (Table 3), which are both bare open surfaces with little or no vegetation. This pattern could be a result of more cloudy days in the two sites, which could have impacted the long-term variation of solar radiation.

The soil moisture loss through the process of evaporation and evapotranspiration is dependent on the prevalent atmospheric condition *e.g.*, wind speed, relative humidity, and

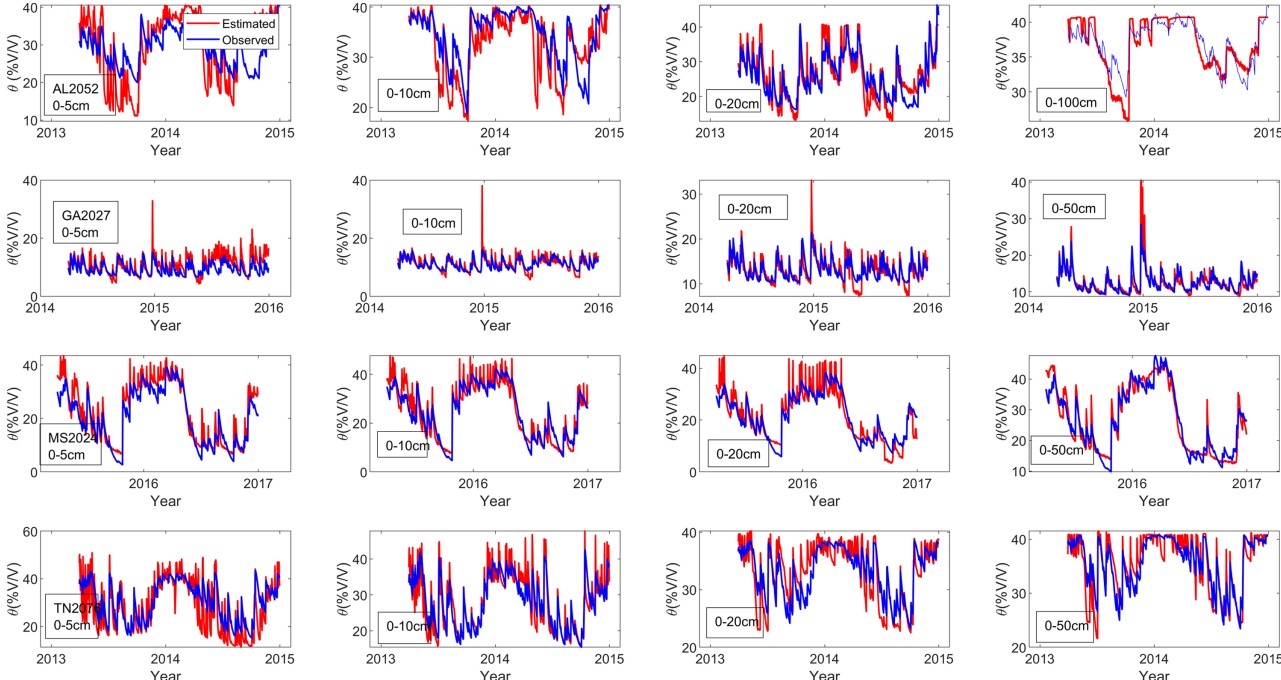

**Figure 4 Estimated and observed soil moisture for columns 0–5, 10, 20, 0–50 and 0–100 cm during both parameter testing periods.**

air temperature (*Pan, 2012*; *Pan, Nieswiadomy & Qian, 2015*). To capture various weather patterns, we incorporated solar radiation data. Also, all the sites studied have limited solar radiation observation data (mostly 5 years). For this research, the estimation of maximum solar radiation was on the long-term average of the actual solar radiation for 5-years. This might not represent the true nature of maximum solar radiation for the sites, thereby affecting the estimation of maximum solar radiation. This can explain the higher RMSEs for sites in the southeastern region. To adequately account for the weather pattern, more years of actual solar radiation is required to enhance the soil moisture loss estimation. In fact, using historical satellite images could help extend solar radiation data at the sites to more than 10 years. However, it is beyond the scope of this article.

The nature of the terrain and slope was not included in the model. Many studies (*e.g.*, *Ridolfi et al., 2003*; *Zhu & Lin, 2011*; *Milledge et al., 2013*; *Traff et al., 2015*) have examined the relationship between the nature of the terrain/slope and soil moisture dynamics. Soil moisture variation is also dependent on the characteristic of the terrain (*e.g.*, slope and aspect). The nature of the terrain may determine if rainfall will infiltrate the ground and increase the soil moisture or lead to surface runoff. This factor was not accounted for in the daily diagnostic equation. This could explain that some of the variations in RMSEs for all the sites examined, but the information about site terrain was not examined. Also, the soil types and properties, such as texture and the depth of the bedrock fragment content, can significantly affect the RMSEs.

Lastly, although solar radiation is an essential element in understanding weather patterns, there are other elements like wind speed, air temperature, and relative humidity

**Table 4 Correlation ($R^2_{\theta,\theta'}$) and RMSEs between estimated and observed soil moisture for model testing period.**

| Site ID | RMSE | $R^2_{\Theta\Theta'}$ |
|---|---|---|
| 5 cm | | |
| AL2053 | 4.47 | 0.8769 |
| GA2027 | 2.86 | 0.7239 |
| MS2025 | 4.32 | 0.8962 |
| TN2076 | 4.40 | 0.7706 |
| 10 cm | | |
| AL2053 | 5.05 | 0.8257 |
| GA2027 | 1.75 | 0.8170 |
| MS2025 | 4.95 | 0.8034 |
| TN2076 | 3.54 | 0.8134 |
| 20 cm | | |
| AL2053 | 4.46 | 0.8757 |
| GA2027 | 2.19 | 0.7999 |
| MS2025 | 6.70 | 0.8747 |
| TN2076 | 2.48 | 0.8471 |
| 50 cm | | |
| GA2027 | 2.69 | 0.7835 |
| MS2025 | 3.79 | 0.8923 |
| TN2076 | 3.44 | 0.7066 |
| AL (0–100 cm) | 4.51 | 0.6130 |

that affect the atmospheric condition of a place. For example, soil moisture loss in a calm climate with low wind speed is lower. Compared to when the wind speed is higher. The relative humidity is the amount of moisture in the atmosphere; its variation will affect the amount of moisture loss from the surface. The relative humidity is also dependent on the air temperature of the location. All these factors control the dynamics of soil moisture loss from any location. In this article, only solar radiation was applied, and there was improvement in soil moisture estimation. The RMSEs slightly reduce and thereby lead to an improvement in the estimation of the soil moisture. This improvement is because more parameters controlling soil moisture dry-down were introduced into the loss coefficient function. To make this model better for future research, other climatic elements should be introduced into the loss function coefficient. Also, to improve the accuracy of soil moisture estimation in vegetated areas, information about the leaf area index (LAI) is important.

Estimating soil moisture using remote sensing is mainly limited to surface soil moisture (*Akbar et al., 2017*; *Kseneman & Gleich, 2013*; *Peng & Loew, 2017*; *Moradizadeh & Saradjian, 2018*; *Chew, Small & Larson, 2016*). Prior studies of soil moisture estimation using the daily diagnostic equation were based on moisture loss using the day of the year as an approximation (*Pan & Nieswiadomy, 2016*; *Pan, Nieswiadomy & Qian, 2015*; *Pan, 2012*). We introduced solar radiation data in the daily diagnostic equation in this study and

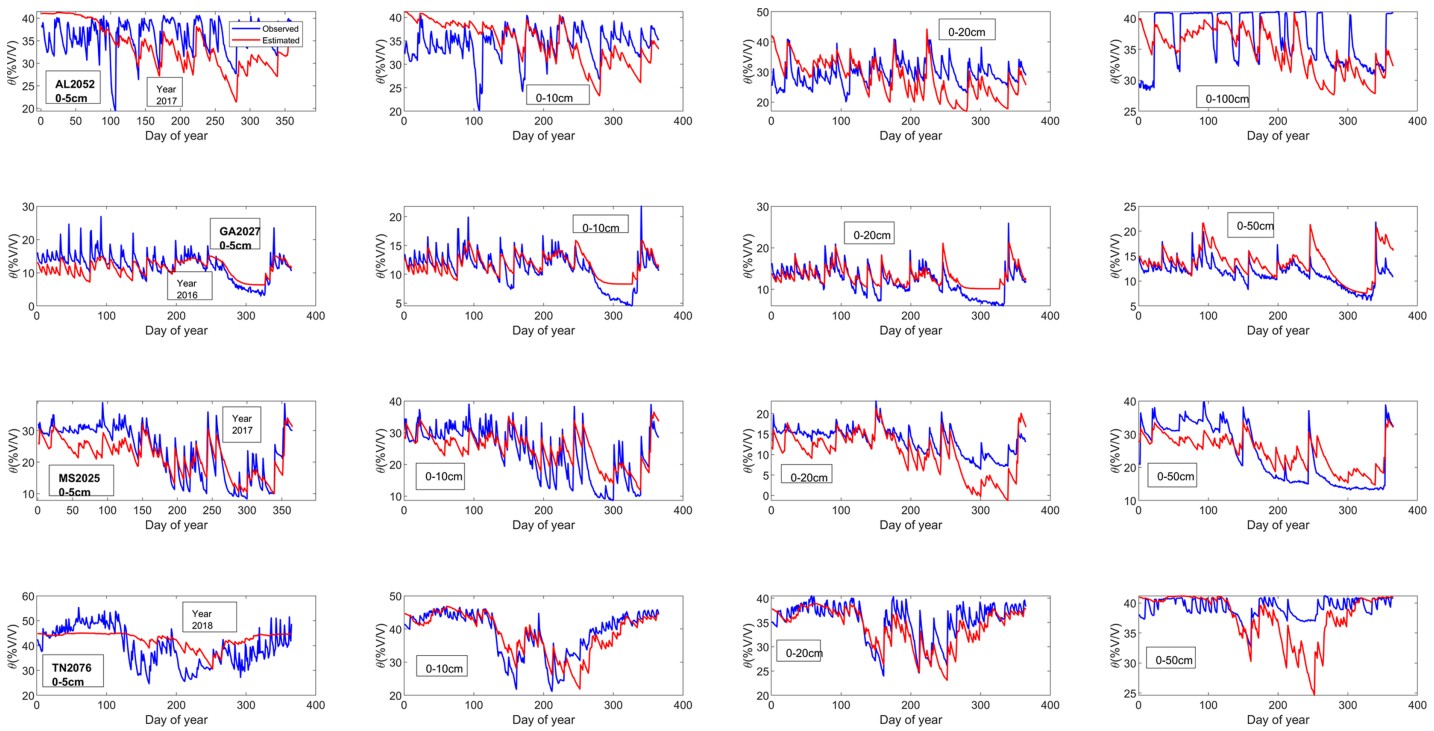

**Figure 5 Estimated and observed soil moisture for columns 0–5, 10, 20, 0–50 and 0–100 cm during both model testing periods.**

observed improved soil moisture estimation for the entire root zone. The experiment showed significant improvement for the topsoil (column 0–5 cm), and this improvement decreased with depth. This study demonstrated the importance of solar radiation data in estimating soil moisture with the diagnostic equation and made a case for including more environmental variables for soil moisture loss in the daily diagnostic equation.

## CONCLUSION

This research proposed an improvement of the daily diagnostic equation, which is the relationship between the loss function coefficient and the summation of historical precipitation. Previously the day of the year (DOY) was used to approximate soil moisture loss through evapotranspiration and evaporation from any location and time, which is the first-order approximation (*Pan, 2012*; *Pan, Nieswiadomy & Qian, 2015*). In this article, two parameters, the clear sky solar radiation, and actual solar radiation, for each day of the year were incorporated into the loss function coefficient. This can be called the second-order approximation. This premise for incorporating solar radiation into the loss function coefficient was based on the concept that solar radiation is the driving force for both atmospheric and biological processes.

The solar radiation parameters were integrated into the loss function coefficient of the soil moisture diagnostic equation (result and discussion section), for the entire root zone (0–5, 0–10, 0–20, 0–50 and 0–100 cm column). The results show improvement at the topsoil (0–5 and 0–10 cm), which was observed to decrease with depth, but there were

some exceptions to the result. A few columns within the soil root zone did not improve after applying solar radiation data. In this study, only 5 years of solar radiation data were used for estimating the clear sky solar radiation, which was incorporated into the loss function coefficient. Our research shows more years of actual solar radiation data are required (at least 10 years) to estimate the clear sky solar radiation and therefore improve the soil moisture estimation using the soil moisture diagnostic equation (SMDE). The SMDE is robust in estimating soil moisture but is based on *in-situ* data for building the soil moisture diagnostic equation model. The limited data undermine the spatial dimension application of this method.

The daily diagnostic equation has been used to estimate soil moisture in arid and semi-arid regions (*Pan, Nieswiadomy & Qian, 2015*). It was also applied to snow-dominated regions (*Pan & Nieswiadomy, 2016*). In this study, actual solar radiation and clear sky solar radiation were included in the sinusoidal wave function to improve soil moisture loss estimation using the daily diagnostic equation. With this approach, we observed improved accuracy of soil moisture estimation. This research has shown that the daily diagnostic equation is robust, simple, and can be improved by applying solar radiation data into its loss function coefficient. Future research on the daily diagnostic equation should focus on the application diagnostic equation to improve crop production and examine the relationship between soil moisture estimation and food security, using the daily diagnostic equation.

## NOTATION

| | |
|---|---|
| $\theta$ | Soil moisture (%(V/V)) |
| $k$ | Soil depth (cm) |
| $C_1$ | Mean of the loss coefficient function (inch or cm/day) |
| $C_2$ | Magnitude of the loss coefficient function (inch or cm/day) |
| $C_3$ | Phase of the loss coefficient function |
| $B$ | B values (dimensionless) |
| Doy | Day of the year |
| $M$ | Precipitation (inch) |
| $\gamma$ | Infiltration rate (Inch/day, mm/day, or cm/day) |

### Funding

The authors received no funding for this work.

### Competing Interests

Lei Wang is an Academic Editor for PeerJ.

## Author Contributions

- Olumide Omotere conceived and designed the experiments, performed the experiments, analyzed the data, prepared figures and/or tables, authored or reviewed drafts of the article, and approved the final draft.
- Feifei Pan conceived and designed the experiments, performed the experiments, analyzed the data, authored or reviewed drafts of the article, and approved the final draft.
- Lei Wang conceived and designed the experiments, authored or reviewed drafts of the article, and approved the final draft.

## Data Availability

The raw data is available in the Supplemental Files.

## Supplemental Information

Supplemental information for this article can be found online at http://dx.doi.org/10.7717/peerj.14561#supplemental-information.

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
