# Peer review of "Using solar radiation data in soil moisture diagnostic equation for estimating root-zone soil moisture"

_PeerJ, doi:10.7717/peerj.14561_

## Round 0.1 · original submission · Major Revisions

Please consider the reviewers' comments carefully!

Reviewer 1 ·

Basic reporting

In this paper, the authors presented an improvement for the soil moisture daily diagnostic equation by introducing some solar parameters. The paper is well-structured and relevant literature is cited. Although the language is generally clear throughout the paper, its form can be improved to be more professional. Some suggestions:
The contract form “it’s” should be replaced with “it is”, (for example lines 108, 109, 189), and “till” (line 173) with “until”, “averagely” (lines 23-25) with “on average”
Some sentences are not immediate to understand. The article could benefit from writing them in a simpler form (eg lines 137-140 “As infiltration diminishes and consequently becomes zero, the value of B increases as the value of soil moisture increases and becomes constant (Pan, Peters-Lidard & Sale, 2003a; Pan Feifei, 2012) because infiltration decreases as B values increase.”)
Awkward repetitions should be avoided (line 136: “The infiltration varies with the soil moisture content, so it cannot be a constant”, line 148 ”soil moisture in the soil”)
Some other minor corrections in form(e.g., line 192 “ϑ and B is (->are) the soil moisture measurement and the computed B values”, line 229 “maybe (->may be)”)
The background of your research is clear and well explained, but some imprecisions in the derivation of the SMDE should be corrected:
Equation 3: the extremes of integration on the left-hand side of the equation should be changed in ϑ_1 and ϑ_2
The length of the soil column is generally indicated by the variable ‘k’, but in Equations 5 and 7 I believe that sometimes it is called ‘z’. If the two variables indicate the same thing, this discrepancy should be corrected.
Equation 7: Double-check the indexes inside the summation. Some of them are probably missing or are incorrect.
Equation 8: a γ in evidence is missing
Equation 16: the expression (ϑ_i+ϑ_i)/2 has to be replaced with (ϑ_i+ϑ_(i-1))/2
Some mistakes when referring to equations: line 103 “equation one (->1) and two (->2)”; line 162 “equation 4 (I think you referred to equation 11 instead)”.
Line 117 “Here, 1, ƞ1 and M1 (->ƞ1, ϑ_1 and M1) are the daily loss coefficient, soil moisture, and precipitation for day 1”
The quality of the figure is satisfactory, but the understanding could be eased by giving additional information.
Figure 1: The label of the y-axis is just the unit of measurement of solar radiation. I suggest writing the name of the variable and in brackets the units of measurement, i.e., “Solar radiation (W/m2)”
Figure 2: It would be useful to specify that the given soil moisture is a volumetric percentage, since it could also be intended as a mass fraction.
Figure 3: It should be explained (at least in the caption) which curve represents the data and which one represents the fit. Additionally, to be unambiguous, the name of the site and the length of the soil column to which this figure refers should be indicated. To conclude, it should be specified that the values on the y axis are the soil moisture values in volumetric percent.

Experimental design

The aim of the research is well defined and is consistent with the scope of the journal. The analysis is performed rigorously. However, the paper could benefit from a deeper explanation of the following aspects.
1. The solar parameters used in the article were calculated from the maximum daily solar radiation of each study site for five years. The details of this calculation are not fully clear in figure 1, specifically:
From Figure 1, it appears that you selected only some points to calculate the fit, and I assume that only the clear-sky days were chosen for this purpose. If this is the case, it would be good to say that in the paper. In addition, the caption of the figure should indicate to which experimental site it refers.
In the figure, only 365 days are shown, whereas data were taken for five years. Do the points represent only the maximum values measured during the five years? Or the plot simply displays one of the years?
2. In figure 2, it appears that for the AL2053 and TN2076 sites soil moisture reaches saturation conditions above certain B values. At first glance, I imagine that this occurs when the pores of the soil are completely filled with water, but looking at the curves and the parameters in Table 3, the estimated effective soil porosity is much higher than the real one. This is quite evident for the AL2053 site for depths of 20 and 100 cm. As you suggested, this might also depend on the fact that the model does not consider some relevant parameters for the site, such as the leaf area index. Could you provide a discussion about this topic? If the porosity of the soil is reasonably known from the soil moisture data, it would be interesting to use this value as ∅_e in equation 10 and fit only the parameters ϑ_res and C4. Then, the fit with two parameters (ϑ_res and C4) and the one with three parameters (∅_e, ϑ_res and C4) could be compared. If the RMSE is quite similar for the two cases, it may be preferable to use the model which contains more accurate information about the soil characteristics.
3. In Table 3, on the last line (AL25023 site, 0-100cm) the calibration parameters are the same for the two forms of the equation. Perhaps that line should be double checked. If the parameters are indeed identical, the reason why should be investigated in detail.

Validity of the findings

The conclusions and developments stated by the authors are well argued, consistent with the purpose of the investigation, and supported by the provided data. I only found some issues with the shared material:
1. The five-year solar data for computing the clear solar radiation are not provided. Is it possible to include them?
2. The MATLAB scripts you provided are quite hard to interpret; I suggest adding more comments to the code and possibly using the same names for the coefficients in the code and the paper.
3. In the script for calculating the ratio between the clear and actual solar radiation, the imported data are multiplied by a factor of 0.48458. Could you give details about the reason why this operation was performed? Is it just a conversion to express the radiation in W/m^2?

Additional comments

The paper “Using solar radiation data in soil moisture diagnostic equation for estimating root-zone soil moisture” presents an improved version of the soil moisture daily diagnostic equation, including some solar parameters in the calculation. The derivation of the equation is provided in the paper, as well as the reasons that justify including the solar parameters in it. Improvement in soil moisture estimation with this new version of the equation is supported by the data in five different fields that are investigated meticulously. The paper is generally clear and the methods are described in detail, but the overall quality could be improved. Most importantly, more details should be given about the estimation of the clear solar radiation. Then, some corrections to the language expressions and double checks to equations, figures, and tables are strongly advised. Sufficient raw data are shared to allow the replication of the analysis performed. I also commend the author for having included the MATLAB code for easing the data analysis.

Reviewer 2 ·

Basic reporting

See attachment

Experimental design

Methods described with sufficient detail & information to replicate. But it needs to be some modification. See attachment.

Validity of the findings

Conclusions are well stated, linked to original research question & limited to supporting results. Please see attached file

Annotated reviews are not available for download in order to protect the identity of reviewers who chose to remain anonymous.

---

## Round 0.2 · Minor Revisions

Overall, the discussion part is weak. The Discussion should summarize the manuscript's main finding(s) in the context of the broader scientific literature and address any study limitations or results that conflict with other published work.

Reviewer 1 ·

Basic reporting

In its revised form, the manuscript appeared clearer and more immediate to understand to me. Some minor remarks could further improve the language quality:
Line 76: soil moisture lose (did you mean ‘loss’?) is dependent (-> depends) on other…
Line 93: The specific research question and contribution of this research is "the" quantification of the improvement
Line 105: θ is "the" soil moisture
Line 114: equation 3; the left-side of the equation should be integrated between θ1 and θ2, instead of p1 and p2
Line 138: “The rainfall contribution to soil moisture from day 1 diminishes due to the decreasing exponential term –[..] in equation 8, which shows the number of days before day 1 increases(Pan, 2012),” although the message is clear from the mathematics, the meaning of the last part of the sentence (“, which shows the number of days before day 1 increases”) is not fully clear to me.
Line 167: “Ƞ denote the soil moisture for the day i”. Isn’t it the loss coefficient?
Line 177: The distinct characteristic of clear sky solar radiation is "that" it increases monotonously…
Line 196: It is assumed that the infiltration coefficient and the loss coefficient for a short time step ( 1 day) is (-> are) independent of time.
Line 232: The correlation coefficient between the observed soil moisture and the B values at columns 0-5cm, 0-10cm, 0-20, 0-50cm, 0-100, show (-> shows) improvement
Line 265: I suggest double checking this sentence: “for the plot of the observed soil moisture versus B values for each column (0-5cm, 0-10cm, 0-20cm, 0-50cm, and 0-100cm) using the least square method (the diagnostic equation (equation 10)) "was used" as the fitting equation (Table 3) (Pan, 2012). The root mean square errors (RMSEs) is (-> are) generally below 4.5%v/v for the entire root zone (compare to (Pan, 2012; Pan et al., 2015a), which was below 5% v/v after (maybe you meant ‘before’?) applying the ratio…

Line 325: could explain some of the variation
Figure 5: some textboxes are too little for the text that is written inside. I suggest that you enlarge them a little.
Table 2: Please double check the parameter in the table. Is it R_(B,θ) or R^2_(B,θ)? If it is the latter, correct the name of the parameter in row 3, columns 5 and 9. Conversely, if it is R_(B,θ), please correct the caption.

Experimental design

My comments on the experimental design were addressed and some procedures explained in greater detail.

Validity of the findings

In this version of the manuscript the purpose of the research and the knowledge gap that the authors want to fill are explicitly stated. Additional data were included in the supplementary material to allow the reproducibility of the work.

Additional comments

The authors revised the paper according to the reviewers suggestions. In this new form, the paper appears clearer and more complete.
If some minor issues are corrected, I advise the manuscript publication.

---

## Round 0.3 · accepted · Accept

I congratulate the authors for the effort put into this paper! The manuscript is significantly improved; therefore, I recommend accepting it in its current form!